# Safety of Interchanging the Live Attenuated MAV/06 Strain and OKA Strain Varicella Vaccines in Children

**DOI:** 10.3390/vaccines11091442

**Published:** 2023-08-31

**Authors:** Hyun Mi Kang, Gwanglok Kim, Young June Choe

**Affiliations:** 1Department of Pediatrics, Seoul St. Mary’s Hospital, College of Medicine, The Catholic University of Korea, Seoul 06591, Republic of Korea; pedhmk@gmail.com; 2Department of Corporate Development, GC Biopharma Corporation, Yongin 16924, Republic of Korea; kimgr@gccorp.com; 3Department of Pediatrics, Korea University College of Medicine, Korea University Anam Hospital, Seoul 02841, Republic of Korea

**Keywords:** MAV/06, safety, vaccine, varicella, OKA

## Abstract

Two live attenuated varicella vaccine (VZV) strains have been mainly used across the globe: MAV/06 and OKA strains. We aimed to explore the safety of interchanging the two VZV strains for primary and booster immunizations. South Korea’s vaccine adverse event reporting system (VAERS) was accessed and searched to find filed reports of all adverse events (AEs) following immunization with the second dose of the varicella vaccine. The electronic medical records were reviewed for all visits to the hospital following the second dose of the varicella vaccine. Of the total 406 study participants, 27.5% (n = 112) were in the MAV/06–MAV/06 group, 30.3% (n = 123) in the MAV/06–OKA, 17.5% (n = 71) in the OKA–MAV/06 group, and 24.6% (n = 100) in the OKA–OKA group. Mean age at immunization with the first dose was 1.10 (standard deviation [SD] ±0.34) years old, and second dose was 4.77 (SD ± 1.13) (*p* = 0.772 and 0.933, respectively). There were no filed reports of AEs following the second dose in the national VAERS. Hospital visit records showed a total of 10.3% (95% confidence interval [CI], 7.6–13.7) (n = 42) had recorded AEs following the 2nd administered dose; however, only 0.7% (95% CI, 0.2–2.4) (n = 3) were regarded as possibly vaccine related. Two patients in the MAV/06–OKA group were diagnosed with Henoch-Schonlein purpura after the second dose; however, both had also received the MMR vaccine on the same day. No safety signals associated with interchanging the MAV/06 and OKA strain live attenuated varicella vaccines were observed in this patient cohort of healthy children.

## 1. Introduction

Interchangeability of vaccines is extremely important because previously administered vaccine products may be unknown or may be inaccessible/unavailable. Furthermore, new vaccines and combination vaccines by different manufacturers are continually emerging, which is also another reason vaccine interchangeability is important. Generally, the same manufacturer’s product is advised for all doses in a vaccine series, however, the lack of availability of a specific product should not defer or delay immunization. Thus, to be considered interchangeable, (1) vaccines must have the same indications with equally acceptable schedules, (2) should contain comparable types and quantities of antigen, and (3) should have the same safety, reactogenicity, immunogenicity and efficacy profiles [1,2].

In 1993, the MAV/06 strain live attenuated varicella vaccine was the world’s second varicella vaccine to be licensed, following BIKEN’s OKA strain live attenuated varicella vaccine, which was licensed in 1974. Since then, the MAV/06 strain varicella vaccine has been incorporated into South Korea’s national immunization program (NIP), and more than 30 million doses have been distributed [3].

In South Korea, the primary dose of the live attenuated varicella vaccine administered at 12 months is included in the NIP. The second booster dose is optional at age 4–6 years old. In the concern of primary vaccine failure and waning immunity [4], awareness on the importance and benefit of the second booster dose is increasing, and more pediatricians and parents have been choosing to administer a second dose to their children. 

There are seven distinct major and one tentative phylogenetic clades (1–6, 9 and VIII) of varicella zoster virus (VZV) [5,6]. Varicella vaccines that are derived from MAV/06 and OKA strains belong in clade 2. In South Korea, children have been immunized with vaccines derived from both strains: MAV/06 strain vaccines including Suduvax (GC Biopharma, Yongin, South Korea) and Barycela (GC Biopharma, Yongin, South Korea), and OKA strain vaccines including Vari-L (Changchun Institute of Biological Products, Jilin, China), SKYVaricella (SK Bioscience, Seongnam-si, South Korea), Varilrix (GlaxoSmithKline, Brentford, United Kingdom), and VARIVAX (Merck, Rahway, NJ, USA). In a postlicensure review on the safety of the licensed varicella vaccines, adverse events reported were considered minor, and serious adverse events were found to be rare [7].

When administering the second dose of the live attenuated varicella vaccine, interchangeability of the two strains (MAV/06 and OKA) has been widely accepted and practiced throughout the country. However, a lack of data exists on the safety of interchanging the two VZV strains from clade 2. Therefore, the purpose of this study was to detect potential hazards of interchanging two different VZV vaccine strains (MAV/06 and OKA) for primary and booster immunizations. 

## 2. Materials and Methods

This was a retrospective cohort study of patients that were 4–9 years old, that were followed up at the infectious disease outpatient clinic during January 2020 to December 2022. The inclusion criteria were as follows: (1) received two doses of the live attenuated varicella vaccine, (2) no underlying immunocompromising diseases, and (3) had records of visitations to Seoul St. Mary’s Hospital’s outpatient clinic or emergency room after the 2nd dose for any reason. The exclusion criteria were as follows: (1) had no recorded vaccination dates, (2) lived abroad at any time after the 2nd dose, (3) had any immunocompromising underlying diseases, and (4) were administered immunosuppressants after the 2nd dose. 

Patients were categorized into a total of four groups depending on the combination of the 1st and 2nd dose of the immunized VZV strain: (A) MAV/06–MAV/06, (B) MAV/06–OKA, (C) OKA–MAV/06, and (D) OKA–OKA group. The total number of patients in each of the groups was based on the proportion of patients immunized with a certain combination in comparison to group D, with group D set at 100 patients. 

For each of the patients, South Korea’s vaccine adverse event reporting system (VAERS) was accessed and searched to find filed reports of all adverse events (AEs) following immunization with the 2nd dose of the varicella vaccine. Furthermore, the electronic medical records were reviewed for all visits to the hospital following the 2nd dose of the varicella vaccine: local AEs (injection site pain, swelling, erythema, cellulitis, rash), systemic AEs (fever, malaise, pneumonia, hepatitis) up to 42 days after immunization. Any events such as herpes zoster, immune-mediated syndromes (including erythema multiforme, SJS, arthropathy, thrombocytopenia, anaphylaxis, vasculitis, aplastic anemia), neurologic syndromes (including neuropathy, convulsion, ataxia, encephalopathy, meningitis) and morality up to 42 days following the 2nd dose of the varicella vaccine were extensively reviewed for each of the patients. The selected AEs were based on AEs reported in the postlicensure safety surveillance for varicella vaccines by the Food and Drug Administration (FDA) [7].

The chi square test was used to compare categorical variables, and the one-way analysis of variance was used to compare continuous variables. All tests were two-sided, and a *p*-value of <0.05 was determined as statistically significant. 

The Institutional Review Boards of Seoul St. Mary’s hospital reviewed and approved this study (IRB no. KC23RISI0425).

## 3. Results

A total of 890 children fit the inclusion criteria, and of these, a total 406 patients were randomly selected to be included as study participants, with 27.5% (n = 112) in the MAV/06–MAV/06 group, 30.3% (n = 123) in the MAV/06–OKA, 17.5% (n = 71) in the OKA–MAV/06 group, and 24.6% (n = 100) in the OKA–OKA group. Mean age at immunization with the first dose was 1.10 (standard deviation [SD] ± 0.34) years old, and the second dose was 4.77 (SD ± 1.13), with no significant difference in the four groups (*p* = 0.772 and 0.933, respectively). The distribution of the administered vaccine products is shown on Table 1. 

Overall, none of the patients included in this study had any filed reports of AEs following the second dose of the live attenuated varicella vaccine in the national VAERS. In the hospital visit record reviews, a total of 10.3% (95% confidence interval [CI], 7.6–13.7) (n = 42) had recorded AEs following the second dose immunization; however, only 0.7% (95% CI, 0.2–2.4) (n = 3) were regarded as possibly vaccine related. No significant differences in the frequency of adverse events were observed according to the vaccine groups (Table 2). 

The most frequently recorded AEs were as follows: lower respiratory tract infections (n = 18; 4.4%; 95% CI, 2.7–6.9), upper respiratory tract infections (n = 9; 2.2%; 95% CI, 1.0–4.2), and gastrointestinal symptoms (n = 4; 1.0%; 95% CI, 0.3–2.5). All were regarded as vaccine-unrelated events. Fever was recorded as the chief complaint for hospital visits in three patients; however, two patients had fever that initiated 4 days after immunization with the second dose, and were concluded as fevers caused by infections and were vaccine unrelated. In one child, fever occurred within 24 h after immunization in the OKA–MAV/06 group, and was regarded as possibly vaccine related. 

A total of three immune-mediated syndromes were diagnosed within 42 days after the second dose; however, one patient was diagnosed with Kawasaki disease, and thus regarded as vaccine unrelated. Two patients in the MAV/06–OKA group were diagnosed and treated for allergic purpura, Henoch-Schonlein purpura (HSP). One patient had symptom onset 11 days after the second dose with an OKA strain vaccine, and another patient 25 days after the second dose of an OKA strain varicella vaccine. Both children had received the MMR vaccine on the same day. 

By vaccine groups, of the three possibly vaccine-related adverse events, two cases of HSP occurred in the MAV/06-OKA group. However, the OKA vaccine product of the second administered dose differed in the two patients (Vari-L and SKYVaricella). The case of vaccine-related fever occurred in the OKA-MAV/06 group, and the patient received the Suduvax, a MAV/06 strain varicella vaccine for the second dose. This patient’s fever began within 24 h after immunization with the second dose of the varicella vaccine, and her fever lasted for two days. No other symptoms or signs of infection were observed, and the patient’s urinalysis was negative (Table 2). 

## 4. Discussion

Both Oka and MAV/06 strain-based varicella vaccines can induce cross-reactive humoral immunity against other clades of VZV [8]. However, no data exist on the safety and tolerability of interchanging the OKA and MAV/06 strain-based vaccines. This study showed that there were no reports of AEs reported to the national VAERS after the second dose of the varicella vaccine in all four groups, and when reviewing hospital visit records, three possibly vaccine-related AEs occurred after the second administered dose of the VZV vaccine, and showed no significant correlation with a specific combination group. 

Allergic purpura, HSP, is a systemic vasculitic syndrome with an unknown etiology [9]. There have been previous reports of HSP following immunization with various vaccines such as COVID-19 mRNA vaccines and influenza vaccines [10,11]. Furthermore, a case–control study in Italy has reported that the measles–mumps–rubella vaccine has been shown to increase the risk of HSP by an odds ratio of 3.4 (95% CI, 1.2–10.0) [12]. Although the varicella vaccine has been shown to trigger HSP, it is extremely rare [13]. Because both children also received that MMR vaccine on the same day as the second dose of varicella vaccine, it is unclear which vaccine triggered HSP. 

The relatively small sample size in comparison to the number of children that received two doses of the live varicella vaccines was a limitation of this study. Furthermore, visits to other hospitals during the period of 42 days post vaccination were not reviewed due to inaccessibility to the data. Finally, each patient was only followed up for up to 42 days after the second dose; therefore, statistical models taking into account the time of observation were difficult to perform. However, in order to overcome this bias, AEs from the national VAERs system was reviewed for all patients, which showed no filed reports. 

To conclude, no safety signals associated with interchanging the MAV/06 and OKA strain live attenuated varicella vaccines were detected in this patient cohort of healthy children. Continuous monitoring of vaccine safety is essential; moreover, this is only a preliminary result in a limited follow-up period and further large scale in-depth analyses must be performed.

## Figures and Tables

**Table 1 vaccines-11-01442-t001:** Demographics and administered vaccines by dose.

			No. of Cases (%)			
	Total	MAV/06-MAV/06	MAV/06-OKA	OKA-MAV/06	OKA-OKA	*p*
	n = 406	n = 112	n = 123	n = 71	n = 100	
Sex, male	207 (60.0)	56 (50.0)	48 (39.02)	39 (55.0)	57 (57.0)	0.276
Age at 1st dose, mean (±SD)	1.10 (±0.34)	1.17 (±0.52)	1.07 (±0.12)	1.06 (±0.09)	1.07 (±0.21)	0.772
Age at 2nd dose, mean (±SD)	4.77 (±1.13)	4.74 (±0.98)	4.83 (±1.21)	4.75 (±1.09)	4.76 (±1.23)	0.933
Underlying disease (%)	5 (1.2)	1 (0.9) ^1^	3 (2.4) ^2^	1 (1.4) ^3^	0	*-*
**Dose 1 vaccine**						
MAV/06						
Suduvax	235 (57.9)	112 (100)	123 (100)			
OKA						
Vari-L	139 (34.2)			53 (74.6)	86 (86.0)	
SKYVaricella	16 (3.9)			5 (7.0)	11 (11.0)	
Varilix	11 (2.7)			9 (12.7)	2 (2.0)	
Varivax	5 (1.2)			4 (5.6)	1 (1.0)	
**Dose 2 vaccine**						
MAV/06						
Suduvax	164 (40.4)	105 (93.8)		59 (83.1)		
Barycela	19 (4.7)	7 (6.3)		12 (16.9)		
OKA						
Vari-L	115 (28.3)		64 (52.0)		51 (51.0)	
SKYVaricella	107 (26.4)		59 (48.0)		48 (48.0)	
VARIVAX	1 (0.2)				1 (1.0)	

Vaccine manufacturer: Suduvax (GC Biopharma, South Korea), Barycela (GC Biopharma, South Korea), Vari-L (Changchun Institute of Biological Products, China), SKYVaricella (SK Bioscience, South Korea), Varilrix (GlaxoSmithKline, United Kingdom), VARIVAX (Merck, USA). ^1^ Type 1 diabetes mellitus (n = 1), ^2^ Tuberous sclerosis (n = 1), atrial septal defect, secundum type (n = 2), ^3^ Type 1 diabetes mellitus (n = 1).

**Table 2 vaccines-11-01442-t002:** Any adverse events reported within 42 days after the second dose of the live attenuated varicella vaccine.

	No. of Cases (%; 95% CI)										
	Total n = 406		MAV/06–MAV/06 n = 112		MAV/06–OKA n = 123		OKA–MAV/06 n = 71		OKA–OKA n = 100		
	Any AE	vAE	Any AE	vAE	Any AE	vAE	Any AE	vAE	Any AE	vAE	*p*
Selected events											
Local reaction	-	-	-	-	-	-	-	-	-	-	*-*
Systemic reaction (fever, myalgia)	3 (0.7; 0.2–2.4)	1 ^1^ (0.2; 0-1.4)	-	-	-	-	2 (2.8; 0.3–9.8)	1 ^1^ (1.4; 0–7.6)	1 (1.0; 0–5.5)	-	0.111
Rash	-	-	-	-	-	-	-	-	-	-	-
Impetigo/cellulitis	1 (0.2; 0–1.4)	-	-	-	1 (0.8; 0–4.5)	-	-	-	-	-	0.511
Hepatic pathology	-	-	-	-	-	-	-	-	-	-	-
URTI	9 (2.2; 1.0–4.2)	-	2 (1.8; 0.2–6.3)	-	2 (1.6; 0.2–5.8)	-	3 (4.2; 0.9–11.9)	-	2 (2.0; 0.2–7.0)	-	0.651
LRTI	18 (4.4; 2.7–6.9)	-	6 (5.4; 2.0–11.3)	-	7 (5.7; 2.3–11.4)	-	1 (1.4; 0–7.6)	-	4 (4.0; 1.1–9.9)	-	0.520
AGE	4 (1.0; 0.3–2.5)	-	1 (0.9; 0–4.9)	-	2 (1.6; 0.2–5.8)	-	-	-	1 (1.0; 0–5.5)	-	0.745
Possible immune-mediated syndromes											
EM	-	-	-	-	-	-	-	-	-	-	-
Arthropathy	-	-	-	-	-	-	-	-	-	-	-
Thrombocytopenia	-	-	-	-	-	-	-	-	-	-	-
Anaphylaxis	-	-	-	-	-	-	-	-	-	-	-
Vasculitis	3 (0.7; 0.2–2.4)	2 (0.5; 0.0–1.8)	-	-	2 (1.6; 0.2–5.8)	2 ^2^ (1.6; 0.2–5.8)	1 ^3^ (1.4; 0–7.6)	-	-	-	0.343
Aplastic anemia	-	-	-	-	-	-	-	-	-	-	-
Neurologic syndromes											
Neuropathy	-	-	-	-	-	-	-	-	-	-	-
Convulsion	1 (0.2; 0–1.4)	-	-	-	1 (0.8; 0–4.5)	-	-	-	-	-	0.511
Encephalopathy	-	-	-	-	-	-	-	-	-	-	-
Meningitis	-	-	-	-	-	-	-	-	-	-	-
Others	3 (0.7; 0.2–2.4)	-	1 ^4^ (0.9; 0–4.9)	-	-	-	-	-	2 ^5^ (2.0; 0.2–7.0)	-	0.302
Total	42 (10.3; 7.6–13.7)	3 (0.7; 0.2–2.4)	10 (8.9; 4.4–15.8)	-	15 (12.2; 7.0–19.3)	2 (1.6; 0.2–5.8)	7 (9.9; 4.1–19.3)	1 (1.4; 0–7.6)	10 (10.0; 4.9–17.6)	-	

^1^ Fever (within 24 h), ^2^ allergic vasculitis (Henoch-Shonlein Purpura), ^3^ Kawasaki disease, ^4^ breast budding, r/o early puberty, ^5^ Amblyopia (n = 1), UTI (n = 1). Abbreviations: AE, adverse events; AGE, acute gastroenteritis; EM, erythema multiforme; LRTI, lower respiratory tract infection; URTI, upper respiratory tract infection; vAE, vaccine-related adverse events.

## Data Availability

Partial data is available upon request.

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
