# Peer review of "Safety of Interchanging the Live Attenuated MAV/06 Strain and OKA Strain Varicella Vaccines in Children"

_vaccines, 2023, doi:10.3390/vaccines11091442_

Round 1

Reviewer 1 Report

Thank you very much let us read and evaluate this interesting brief report (Safety of interchanging the live attenuated MAV/06 strain and vOKA strain varicella vaccines in children), although it is a brief report (and national wise) but releases very interesting results on a narrow point the immunity and durability inducing by interchanged primary and secondary doses of vaccines (MAV/06 and vOKA), which should increase the international awareness with this point. the MS is distinguished by a very clear experimental design. it is an acceptable version just needs minor additions, which seem important to impact the idea.

minor comments:

1. authors encourage to give a brief historical view on interchangable vaccines immunization.

2. authors have 890 children who fit the inclusion criteria they put, but they select 406, why?

3. could you please add this (890 versus 406) in the material and methods section?

Author Response

Reviewer 1) Thank you very much let us read and evaluate this interesting brief report (Safety of interchanging the live attenuated MAV/06 strain and vOKA strain varicella vaccines in children), although it is a brief report (and national wise) but releases very interesting results on a narrow point the immunity and durability inducing by interchanged primary and secondary doses of vaccines (MAV/06 and vOKA), which should increase the international awareness with this point. the MS is distinguished by a very clear experimental design. it is an acceptable version just needs minor additions, which seem important to impact the idea.

--> Thank you very much for the positive and constructive comments.

Minor comments:

  1. authors encourage to give a brief historical view on interchangable vaccines immunization.

--> We appreciated this comment. We have included information on the historical view on interchangeable vaccines immunization as follows:

Introduction section: “Interchangeability of vaccines is extremely important because previously administered vaccine products may be unknown or may be inaccessible/unavailable. Furthermore, new vaccines and combination vaccines by different manufacturers are continually emerging, which is also another reason vaccine interchangeability is important. Generally, the same manufacturer's product is advised for all doses in a vaccine series, however, the lack of availability of a specific product should not defer or delay immunization. Thus, to be considered interchangeable, 1) vaccines must have the same indications with equally acceptable schedules, 2) should contain comparable type and quantity of antigen, and 3) should have the same safety, reactogenicity, immunogenicity and efficacy profiles [1, 2].”

2. authors have 890 children who fit the inclusion criteria they put, but they select 406, why?

--> Thank you for this question. Our aim was to review a representative sample of the entire population, therefore, our aim was to review about 100 patients in each cohort.

3. could you please add this (890 versus 406) in the material and methods section?

--> We appreciate the suggestion and have included the following:

Methods section: “The total number of patients in each of the groups was based on the proportion of pa-tients immunized with the combination in comparison to group D (vOKA-vOKA), with group D set at 100 patients, in order to review a representative sample of the en-tire population. Thus, during the three-year period, a total of 890 children fit the inclu-sion criteria, and 406 were included.”

Reviewer 2 Report

I was invited to revise the paper entitled "Safety of interchanging the live attenuated MAV/06 strain and vOKA strain varicella vaccines in children". It was a retrospective cohort study aimed to evaluate the interchangebility of two different varicella zoster vaccine strains (MAV/06 and vOKA) for primary and booster immunizations.

The topic is interesting and can improve the knowledge on this field.

Despite that, I have several observations:

- Introduction section was poor and needs improvements;

- It was presented as a cohort study. How many person/months were observed?

- Statistical analysis was poor. Authors should perform statisitcal models able to tacke into accoutn the time of the observation;

- In table 1 more patients characteristics should be reported;

- Discussion section was too poor and need some improvements, comparing also similar studies.

Author Response

I was invited to revise the paper entitled "Safety of interchanging the live attenuated MAV/06 strain and vOKA strain varicella vaccines in children". It was a retrospective cohort study aimed to evaluate the interchangebility of two different varicella zoster vaccine strains (MAV/06 and vOKA) for primary and booster immunizations.

The topic is interesting and can improve the knowledge on this field.

--> Thank you for the comments and review.

Despite that, I have several observations:

1) Introduction section was poor and needs improvements;

--> We appreciate this comment. We included the following in the introduction:

Introduction section: “Interchangeability of vaccines is extremely important because previously administered vaccine products may be unknown or may be inaccessible/unavailable. Furthermore, new vaccines and combination vaccines by different manufacturers are continually emerging, which is also another reason vaccine interchangeability is important. Generally, the same manufacturer's product is advised for all doses in a vaccine series, however, the lack of availability of a specific product should not defer or delay immunization. Thus, to be considered interchangeable, 1) vaccines must have the same indications with equally acceptable schedules, 2) should contain comparable type and quantity of antigen, and 3) should have the same safety, reactogenicity, immunogenicity and efficacy profiles [1, 2].”

2) It was presented as a cohort study. How many person/months were observed?

--> Thank you for this question. This was a cohort study of patients of 406 patients that received two doses of the live attenuated varicella vaccine that fit the inclusion criteria during a 36-month period. However, we are unable to calculate person/months because for each patient, all adverse events up to 42 days were recorded, and further follow up was not reviewed.

3) Statistical analysis was poor. Authors should perform statisitcal models able to tacke into accoutn the time of the observation;

--> Again, we appreciate this important comment, however, each patient was only followed up for up to 42 days after the 2nd dose, therefore, statistical models taking into account the time of observation was difficult to perform. The FDA also did not perform such statistical models in their study of adverse events occurring after varicella vaccination (‘Postlicensure safety surveillance for varicella vaccine.’ Jama 2000, 284, (10), 1271-9.).

4) In table 1 more patients characteristics should be reported;

--> Thank you for the suggestion. Because we excluded children with immunocompromising underlying diseases, and all the children were immunized according to the indications for authorized varicella vaccines, we did not consider any other patient characteristics to be important. However, we agree and as advised, we reviewed non-immunocompromising underlying diseases, and included them in Table 1.

5) Discussion section was too poor and need some improvements, comparing also similar studies.

--> Thank you for the recommendation. This is the first study on the interchangeability of two different varicella vaccines, therefore it was impossible to compare similar studies. We included this in the conclusion. Furthermore, because of the word limit that a brief report has compared to an original study, we were unable to discuss further.

Discussion section: “To conclude, this was the first study conducted to observe any adverse events related to interchanging the vOKA and MAV/06 strain-based vaccines. No safety signals associated with interchanging the MAV/06 and OKA strain live attenuated varicella vaccines were detected in this patient cohort of healthy children. Continuous monitoring of vaccine safety is essential.”

Reviewer 3 Report

This is an interesting paper though basically meaningless.

There have been adverse safety reviews published for vOka in the early days of vaccination. Those papers are not referenced or compared against. They should be understood at the very least. 

In Table I, anywhere that is simple counting of patients there is no need for p-values or any other type of statistics. I completely understand that the ages of first and second vaccinations need some statistics. 

Table II is the heart of the paper and I appreciate that the authors broke out the different reactivations to the vaccinations. 

In Table II, the table descriptor ("caes" should be "cases") is misspelled and the registration of the table is a little weird. Again when simply counting patients, no statistics are necessary. 

Vaccines are not just the live virus but also include adjuvants. I would guess most of the reactions to the vaccination were due to the adjuvant rather than the live virus. It would be unconscious able to give patients just adjuvants. 

Again, suduvax and varivax are virtually identical genetically so it not surprising that they can substituted freely.  

Their English is basically okay. 

Author Response

1) This is an interesting paper though basically meaningless.

--> Thank you for taking the time to review this paper. We agree that this can possibly be meaningless because it is a live vaccine. However, we also disagree because interchangeability of vaccines is extremely important because previously administered vaccine products may be unknown or may be inaccessible/unavailable. However, there is currently no data supporting the safety of interchangeability of the two different strains of varicella vaccines, and has been carried out without assessing possible adverse events regarding interchanging the two different strains.

We included this in the introduction section.

"Interchangeability of vaccines is extremely important because previously administered vaccine products may be unknown or may be inaccessible/unavailable. Furthermore, new vaccines and combination vaccines by different manufacturers are continually emerging, which is also another reason vaccine interchangeability is important. Generally, the same manufacturer's product is advised for all doses in a vaccine series, however, the lack of availability of a specific product should not defer or delay immunization. Thus, to be considered interchangeable, 1) vaccines must have the same indications with equally acceptable schedules, 2) should contain comparable type and quantity of antigen, and 3) should have the same safety, reactogenicity, immunogenicity and efficacy profiles [1, 2]."

2) There have been adverse safety reviews published for vOka in the early days of vaccination. Those papers are not referenced or compared against. They should be understood at the very least.

--> The post-licensure safety data from JAMA (JAMA. 2000;284(10):1271-1279. doi:10.1001/jama.284.10.1271) was referenced, however not in the introduction or discussion section, but the methods section. Thank you for this comment, and we included this in the introduction section:

" In a postlicensure review on the safety of the licensed varicella vaccines, adverse events reported were considered minor, and serious adverse events were found to be rare [7]."

3) In Table I, anywhere that is simple counting of patients there is no need for p-values or any other type of statistics. I completely understand that the ages of first and second vaccinations need some statistics.

--> Thank you for the suggestion. We did not include P for dose 1 and 2 vaccines, but we did for sex, because we wanted to show that there were no biases in sex between the groups.

4) Table II is the heart of the paper and I appreciate that the authors broke out the different reactivations to the vaccinations.

--> Thank you for this comment.

5) In Table II, the table descriptor ("caes" should be "cases") is misspelled and the registration of the table is a little weird. Again when simply counting patients, no statistics are necessary.

--> Thank you for the correction.

--> We wanted to show that the frequency of adverse events in each group was not statistically significant.

6) Vaccines are not just the live virus but also include adjuvants. I would guess most of the reactions to the vaccination were due to the adjuvant rather than the live virus. It would be unconscious able to give patients just adjuvants.

--> We do not believe any adjuvants are contained in the live attenuated vaccines (reference: https://www.cdc.gov/vaccines/pubs/pinkbook/downloads/varicella.pdf). We do agree that in vaccines with adjuvants, many adverse reactions and reactogenicity results from adjuvants.

6) Again, suduvax and varivax are virtually identical genetically so it not surprising that they can substituted freely. 

--> We agree, however, again there is a lack to published data to support this, which is why we believe it is important to publish this negative data.

Round 2

Reviewer 2 Report

Authors did not addressed previous comments:

- If all patients were followed for 36 months, adverse events occurred prior the end of each followup so p/t should be assessed.

The presented analysis was poor and limited. 

Author Response

Dear reviewer, 

Thank you for the review. I am sorry that we have not made it clear to you. As we have addressed in the last revision, patients were followed up to 42 days after the 2nd dose. The 36-month period is the study duration in which patients were enrolled. 

We agree that there were limitations in this study, which we have addressed as advised in the limitations section as follows:

** Limitations section: "Finally, each patient was only followed up for up to 42 days after the 2nd dose, therefore, statistical models taking into account the time of observation was difficult to perform. "

However, we ask for you consideration on this point because this is a brief report or preliminary analyses on a representative population. As advised, we plan to do a patient/time analyses, however, in this cohort, only adverse events within 42 days were considered. This 42 day time frame was the same time frame used in clinical phase 3 trials for varicella vaccines to assess vaccine-related adverse events. 

Thank you for your review. 

Round 3

Reviewer 2 Report

Authors should highlight in discussion that this is only a preliminary result in a limited followup period and a deep analysis in the full report has to be performed.

Author Response

Dear Reviewer, 

Authors should highlight in discussion that this is only a preliminary result in a limited followup period and a deep analysis in the full report has to be performed.

--> We, the authors, thank you very much for the important suggestion. We have included the statement in which you have recommended in our discussion section as follows: 

"Continuous monitoring of vaccine safety is essential, moreover, this is only a preliminary result in a limited follow-up period and further large scale in-depth analyses must be performed."

Again, we sincerely appreciate your time to enhance the quality of our article. 

Sincerely, 

Hyun Mi Kang, Gwanglok Kim, and Young June Choe.